# Towards Higher Ranks via Adversarial Weight Pruning

**Yuchuan Tian**[1], **Hanting Chen**[2], **Tianyu Guo**[2], **Chao Xu**[1], **Yunhe Wang**[2]*

[1] National Key Lab of General AI, School of Intelligence Science and Technology, Peking University.
[2] Huawei Noah's Ark Lab.
`tianyc@stu.pku.edu.cn`, `{chenhanting,tianyu.guo,yunhe.wang}@huawei.com`,
`xuchao@cis.pku.edu.cn`

## Abstract

Convolutional Neural Networks (CNNs) are hard to deploy on edge devices due to its high computation and storage complexities. As a common practice for model compression, network pruning consists of two major categories: unstructured and structured pruning, where unstructured pruning constantly performs better. However, unstructured pruning presents a structured pattern at high pruning rates, which limits its performance. To this end, we propose a Rank-based PruninG (RPG) method to maintain the ranks of sparse weights in an adversarial manner. In each step, we minimize the low-rank approximation error for the weight matrices using singular value decomposition, and maximize their distance by pushing the weight matrices away from its low rank approximation. This rank-based optimization objective guides sparse weights towards a high-rank topology. The proposed method is conducted in a gradual pruning fashion to stabilize the change of rank during training. Experimental results on various datasets and different tasks demonstrate the effectiveness of our algorithm in high sparsity. The proposed RPG outperforms the state-of-the-art performance by 1.13% top-1 accuracy on ImageNet in ResNet-50 with 98% sparsity. The codes are available at `https://github.com/huawei-noah/Efficient-Computing/tree/master/Pruning/RPG` and `https://gitee.com/mindspore/models/tree/master/research/cv/RPG`.

## 1 Introduction

As Convolutional Neural Networks (CNNs) are adapted to various tasks at better performance, their sizes also explode accordingly. From shallow CNNs like LeNet [30], larger CNNs like AlexNet [27], to deeper modern CNNs like ResNets [22] and DenseNets [25], CNNs are growing larger for more complex tasks and representations, including large-scale image classification and downstream tasks like object detection [42], segmentation [21], etc. The evolution of CNN gives rise to various real-world applications, such as autonomous driving [2], camera image processing [3], optical character recognition [46], and facial recognition [50]. However, it is difficult to deploy large CNN models on mobile devices since they require heavy storage and computation. For example, deploying a ResNet-50 [22] model costs 8.2G FLOPs for processing a single image with $224 \times 224$ size, which is unaffordable for edge devices with limited computing power such as cellphones and drones.

In order to compress heavy deep models, various methodologies have been proposed, including Weight quantization [19, 56], knowledge distillation [24, 44], and network pruning. Network pruning prunes the redundant weights in convolutional neural networks to shrink models. Weight (or unstructured) pruning [20] and filter (or structured) pruning [32] are two main pathways to prune CNNs. Weight

---

*Corresponding Author.

pruning sparsifies dense kernel weight tensors in convolutional layers in an unstructured manner including iterative pruning [20], gradual pruning [60, 17], and iterative rewinding [15, 16, 43]. Some other works [31, 53, 48] propose gradient or hessian based weight saliencies that proved effective in certain scenarios. Filter pruning [37, 23, 35, 36, 51] prunes filters in convolutional layers as a whole, reducing the redundant width of network layers.

Although structured pruning algorithms can be well supported by existing hardwares and bring large runtime acceleration benefits, their performance is much lower than that of unstructured pruning. For example, SOTA unstructured pruning methods could achieve 80% sparsity on ResNet-50 with little performance drop [48, 45] while structured pruning could only reach less than 50% [49], since filter pruning is a subset of weight pruning by further imposing structural constraints. However, under circumstances of high sparsities, we observe that unstructured pruning partially degrade to structured pruning. When weights are with a large proportion of zeros, it is highly likely that a structured pattern appears, where a whole channel or filter is almost completely pruned. Therefore, existing weight pruning methods usually meet dramatic performance decay at high sparsities.

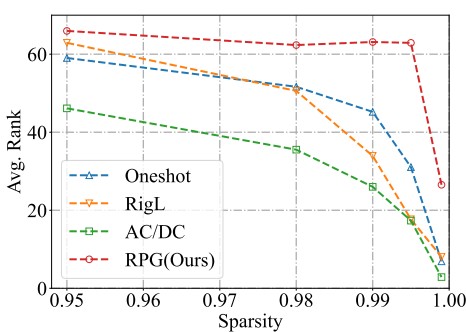

Figure 1. Average weight matrix rank of ResNet-32 [22] pruning baselines versus sparsity. Our rank-based method is effective in maintaining weight ranks at high sparsities compared with baselines.

Inspired by the comparison of the two pruning categories, we propose to reduce structural patterns in weight pruning. Structured pruning is factually a reduction of weight rank in deep Convnets. Thus, rank could be adopted as a metric for evaluating the "structuredness" of unstructured sparse weights: a sparse weight is considered highly structured if it possesses low rank. To keep unstructured pruning from being too structured, we hope to maintain weight ranks under high sparsities in pruning. Based on the goal of rank improvement, we propose an adversarial Rank-based PruninG (RPG) approach for unstructured pruning. First, we find a low-rank approximation of the weight by minimizing the approximation error. The best low-rank approximation is found via singular value decomposition. Second, to enhance weight ranks, we maximize the distance between the weight and its low-rank counterpart to increase weight rank. This adversarial rank-based optimization objective guides sparse weights towards a high-

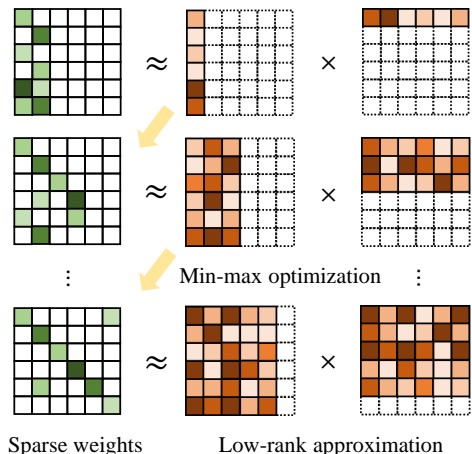

Figure 2. An illustrative diagram of our Rank-based Pruning (RPG) method.

rank topology. The proposed method is conducted in a gradual pruning fashion to stabilize the change of rank during training. The advantage of the proposed RPG method is evaluated through extensive experiments on image classification and downstream tasks, and Figure 1 demonstrates that our method gains a matrix rank advantage compared to baselines.

## 2 Maintaining Rank via Adversarial Pruning

### 2.1 Problem Formulation

In conventional terms of supervised neural network learning, given a target loss function $\mathcal{L}$, neural network weight $W$, and input output pairs $X = \{x_i\}_{i=1...n}$, $Y = \{y_i\}_{i=1...n}$, weight training of a

neural network $W$ is formulated as:

$$\underset{W}{\arg\min}\, \mathcal{L}(Y, WX), \tag{2.1}$$

Weight pruning limits the total number of non-zero weights in weight $W$; or mathematically, weight pruning imposes a $l_0$-norm constraint on neural network learning. Given sparsity budget $c$, the constraint is described as:

$$\|W\|_0 \leq c, \tag{2.2}$$

A common practice is to reparameterize weight $W$ with the Hadamard elementwise product of a weight tensor $W$ and a binary mask $M$. The binary mask $M$ has the same shape as $W$, and each element in $M$ represents whether its corresponding parameter in $W$ is pruned. After reparametrization, the weight pruning problem is then formulated as:

$$\underset{W \odot M}{\arg\min}\, \mathcal{L}(Y, (W \odot M)X) \text{ s.t. } \|M\|_0 \leq c. \tag{2.3}$$

In Equation (2.3), $\odot$ is the Hadamard elementwise product of matrices.

At high sparsities in unstructured pruning, the rank of sparse networks could decrease substantially. In the following sections, we will demonstrate the problem and propose a solution to maintain sparse weight ranks.

## 2.2 Analyzing Weight Pruning in High Sparsity

Unstructured and structured pruning are two major pruning methodologies. In unstructured pruning practices, weight tensors of CNNs are pruned in a fine-grained manner: each and every solitary weight parameters could be turned off (*i.e.* set to zero) within the network, but the whole weight tensor structure is left unchanged. In contrast, structured pruning focuses on the pruning of filters: filters are cut-off as the smallest prunable unit in the pruning process. Comparing the two pruning paradigms under the same sparsity budget, Zhu and Gupta [60] illustrate that unstructured pruning performs much better than structured pruning under the same pruning budget.

This phenomenon could be explained from the perspective of matrix ranks. In fact, structured pruning is a direct rank reduce imposed on weight matrices, which means filter pruning is basically weight pruning with low rank. The rank of a matrix represents the upper bound of the amount of information contained in the matrix. A powerful network should be rich in information, and we hope features of the sparse network could have high ranks. Feature ranks is closely related to ranks of sparse weight matrices because of the formula below that describes the relationship of ranks in matrix multiplication:

$$\text{Rank}(Y) = \text{Rank}(WX) \leq \min\big(\text{Rank}(W), \text{Rank}(X)\big). \tag{2.4}$$

According to Equation (2.4), when filter pruning is applied on weight $W$ that directly impacts its rank, the rank of the output feature will also degrade, causing dramatic loss in information richness. On the other hand, unstructured pruning is free from the structural constraint of filter pruning, and thus maintain more amount of information.

However, under circumstances of high sparsities, we observe that unstructured pruning partially degrades to structured pruning. When weights are filled with a large proportion of zeros, it is very probably that some filters or channels are almost entirely turned-off: "quasi-structured" sparse weight pattern is then formed. A baseline evaluation of matrix ranks in Figure 1 illustrates this concern. Therefore, existing weight pruning methods usually meet dramatic performance decay at high sparsities. Inspired by the properties of the two categories of pruning, we propose to reduce the structured pattern in unstructured pruning, and therefore to maintain weight ranks under high sparsities.

## 2.3 Low Rank Approximation through SVD

Now that weight ranks are important in weight pruning, we need a practical way to compute ranks in the context of deep neural networks. Previous deep learning works on ranks apply matrix rank theories to CNN low-rank tensor decomposition. In these works, low-rank approximations are proposed to fit

weight tensors. Denil et al. [10] decomposites $W \in \mathbb{R}^{m \times n}$ into the multiplication of $U \in \mathbb{R}^{m \times r}$ and $V \in \mathbb{R}^{r \times n}$ where $r$ is much smaller than $m$ and $n$. $UV$ provides a low-rank approximation (rank bounded by $r$) of weight $W$. Denton et al. [11] uses the sum of $k$ rank-one approximations to provide the $k$-rank approximation of a feature tensor. Zhang et al. [57] multiplies a low-rank matrix $M$ to weight matrix $W$, and solves the low-rank $M$ with Singular Value Decomposition (SVD). In modern works [41, 7], low-rank approximations are widely studied as well.

Since the weight values are always discrete, as an alternative solution and inspired by low-rank approximation works, we converge to an approximated rank rather than compute a precise rank solution. Hence, we define the approximated rank as following:

**Definition 1** ($\delta$-rank of a matrix). *Given a matrix $W$ and a small error tolarance $\delta > 0$, the $\delta$-rank of $W$ is defined as the smallest positive integer $k$ such that there exist a $k$-rank matrix, whose $l_2$ distance to $W$ is smaller than $\delta$.*

In previous works, ranks are evaluated via singular values computed from Singular Value Decomposition (SVD). Zhang et al. [57] uses the sum of the top-k PCA eigenvalues to approximate ranks of layer responses; Lin et al. [33] defines rank as the number of non-negligible singular values and does SVD analysis on feature maps; Shu et al. [47] performs SVD on attention maps and augment model performance by keeping a fatter tail in the singular value distribution. These discoveries all acknowledge that singular values from SVD estimates ranks of matrices. We also leverage SVD to compute $\delta$-rank as defined in Definition 1. First, we illustrate that SVD could generate the best low-rank approximation:

**Theorem 1** (The best low-rank approximation). *Suppose $W$ is decomposed via SVD and yield $W = \sum_{i=1}^{r} \sigma_i u_i v_i^T$ where singular values $\{\sigma_i\}$ are sorted in descending order. Given integer $k < r$, the best $k$-rank approximation of $W$, namely the $k$-rank matrix that has the smallest $l_2$ distance to $W$ is*

$$\widetilde{W} = \sum_{i=1}^{k} \sigma_i u_i v_i^T.$$

The proof of Theorem 1 will be shown in Appendix. Since SVD could yield the best low-rank approximation, we could use this property to solve $\delta$-rank defined in Definition 1. Given weight matrix $W$, we search for the smallest $k$ such that the $l_2$ approximation error of best $k$-rank approximation $\widetilde{W}$ as formulated in Theorem 1 is below the error tolerance $\delta$. In this way, we are able to solve the rank of $W$.

## 2.4 Adversarial Optimization for Rank Maintenance

Equipped with the method for matrix rank computation, we hope to formulate a target loss function according to this heuristic such that optimization of the loss could maintain weight ranks.

In contrast to low-rank approximations, high-rank matrices should be hard for low-rank matrices to approximate. Assume $S$ is the set of all low-rank matrices, $W$ should keep its distance away from this set $S$ to increase its rank. But this is a hard problem, for we have to figure out all low-rank matrices. To further simplify the problem, we find the best low-rank approximation rather than all low-rank approximations. $W$ should estrange itself from the best low-rank approximation whose distance is the farthest from $W$. This simplification is valid and will be proved later.

Using this heuristic as motivation, we design an adversarial mechanism that increase the difficulty for $W$ to be approximated by low-rank matrices, and consequently to advocate higher matrix ranks of $W$ while pruning. At first, the best low-rank approximation $\widetilde{W}$ of a small rank $k$ is generated via Singular Value Decomposition, for the purpose of minimizing its distance to weight $W$; next, $W$ is optimized to increase the distance from $\widetilde{W}$. The procedures could be understood as an adversarial combat between $W$ and $\widetilde{W}$: as the low-rank $\widetilde{W}$ tries to fit $W$, $W$ is optimized to keep itself far away from $\widetilde{W}$. Mathematically, the combat could be expressed as a min-max problem.

But unluckily, the problem may suffer the risk of not getting converged. When $\widetilde{W}$ is fixed, the best $W$ is taken when $W \to \infty$. To resolve this issue during optimization, we constrain $W$ within a euclidean norm ball. In other words, we plug $\frac{W}{\|W\|_F}$ instead of $W$ into the max-min problem. The reasons we use $l_2$ normalization are: 1. $W$ is bounded rather than growing to infinity; 2. the rank of

$W$ could increase if we $l_2$ normalize $W$ when optimizing the min-max problem, which will be shown in the mathematical proof in the appendix; 3. $l_2$ normalization on weight is equivalent to imposing $l_2$ normalization on its singular values, providing a fair standard for rank comparisons based on the definition of rank in Definition 1 given fixed error tolerance.

Before the introduction of this min-max problem, we introduce several notations: $\| \cdot \|_F$ is the Frobenius norm (2-norm) of matrices; $I$ is the identity matrix; $\overline{W} := \frac{W}{\|W\|}$ is the $l_2$ normalized weight matrix $W$; $U, \Sigma, V$ are matrices reached from the SVD of $\overline{W}$, where $U = \{u_1, u_2, ...\}$ and $V = \{v_1, v_2, ...\}$ are orthonormal bases; $\Sigma$ is a diagonal matrix where singular values $\{\sigma_1, \sigma_2, ...\}$ are sorted in descending order on the diagonal; operator $\text{Trun}\left(U\Sigma V^T\right) = \sum_{i=1}^{k} \sigma_i u_i v_i^T$ stands for $k$-rank truncated SVD, or the $k$-rank best approximation of $\overline{W}$.

Then formally, we express the min-max problem as follows:

$$\min_{W} \max_{U,\Sigma,V} -\|\overline{W} - \text{Trun}\left(U\Sigma V^T\right)\|_F^2,$$
$$\text{s.t.} \quad U^T U = I, \quad V^T V = I, \quad \overline{W} = \frac{W}{\|W\|}. \tag{2.5}$$

The optimization target is defined as the adversarial rank loss:

$$\mathcal{L}_{rank} = -\|\overline{W} - \text{Trun}\left(U\Sigma V^T\right)\|_F^2. \tag{2.6}$$

In deep learning, gradient descent is the most widely applied method for optimization problems, and we also adopt gradient descent for our experiments. Hence in this context, we propose the following theorem, stating that our adversarial rank loss could guide weight $W$ towards higher rank:

**Theorem 2** (Effectiveness of the adversarial rank loss). *Given the adversarial rank loss as defined in Equation* (2.6). *If we optimize $W$ in rank loss via gradient descent, the rank of $W$ will increase.*

The theorem could be mathematically proved, and the detailed proof will be provided in the appendix.

With the proposed adversarial rank loss, our optimization objective consists of two goals: 1. we hope to reduce the loss for a certain task (*e.g.* classification, detection, etc.) for good sparse network performance; 2. we hope to reduce rank loss for higher weight ranks. We formulate the Rank-based Pruning objective by doing affine combination of the two goals. Given affine hyperparmeter $\lambda$, the loss for a certain task $\mathcal{L}_{task}$, the adversarial rank loss $\mathcal{L}_{rank}$, the Rank-based Pruning (RPG) objective $\mathcal{L}$ is defined as:

$$\mathcal{L} := \mathcal{L}_{task} + \lambda \mathcal{L}_{rank}. \tag{2.7}$$

## 2.5 The Gradual Pruning Framework

Previous works have proposed various pruning framework, including One-shot Pruning [20], Sparse-to-sparse Pruning [8, 1, 39, 12, 14], and Iterative Magnitude Pruning for Lottery Tickets [15, 43]. Compared with these frameworks, Gradual Pruning (GP) [60] could reach better performance with modest training budget. We adopt Gradual Pruning as the pruning framework, which is a usual practice in many works [48, 59, 34]. GP prunes a small portion of weights once every $\Delta T$ training steps, trying to maintain sparse network performance via iterative "pruning and training" procedures.

However, it is hard to associate rank loss with Gradual Pruning; we hope the factor of rank could be considered in the choice of weights via the proposed rank loss. Loss gradients are widely-applied weight saliency criteria, because gradient magnitudes reflect the potential importance of pruned weights: if a turned-off weight possesses large gradients with respect to the objective loss function, it is expected for significant contributions to loss reduction [14]. We use periodic gradient-based weight grow similar to previous pruning works [14, 34, 6], i.e. the weights are periodicly grown at each binary mask update step. But differently, the rank-based pruning objective (defined as Equation (2.7)) is used for gradients computation with respect to each model weight in our case. In this way, the rank factor is considered during the selection of active weights: there is a tendency that RPG chooses an active set of weights that features high-rank.

| Models | VGG19 | | | ResNet32 | | |
|---|---|---|---|---|---|---|
| Sparsity | 99% | 99.5% | 99.9% | 99% | 99.5% | 99.9% |
| Dense | 93.84 | | | 94.78 | | |
| PBW [19] | 90.89 | 10.00 | 10.00 | 77.03 | 73.03 | 38.64 |
| MLPrune [55] | 91.44 | 88.18 | 65.38 | 76.88 | 67.66 | 36.09 |
| ProbMask [59] | 93.38 | 92.65 | 89.79 | 91.79 | 89.34 | 76.87 |
| AC/DC [40] | 93.35 | 80.38 | 78.91 | **91.97** | 88.91 | 85.07 |
| RPG (Ours) | **93.62** | **93.13** | **90.49** | 91.61 | **91.14** | **89.36** |

Table 1. Sparsified VGG-19 and ResNet-32 on CIFAR-10. Baseline results are obtained from [59].

An embedded benefit of periodic gradient-based weight grow lies in computation cost considerations. Singular Value Decomposition (SVD) that is essential for rank computation is costly for large weight tensors. Calculating rank loss for each optimization step is hardly affordable. The adoption of periodic weight updating, however, amortizes the cost of rank loss computations. We also provide an SVD overhead analysis in Sec 3.6.

In summary, Our Rank-based Pruning (RPG) method is formulated as follows: once every $\Delta T$ training steps, the prune-and-grow procedures that updates binary mask $M$ is performed. Firstly, we plan the number of parameters to prune and to grow, such that after mask updating, the whole network will reach the target sparsity at the current iteration. Target sparsity will increase gradually as training goes on, which is identical to GP. Secondly, we globally sort all parameters based on magnitude and perform the pruning operation. Thirdly, we grow the parameters based on gradient. For other training steps, mask $M$ is left unchanged; the active weight values are updated.

Specifically, HRank [33] also leverages matrix rank evaluations in pruning. Our idea is significantly different from HRank [33] in the following aspects: 1. HRank performs filter pruning while our work focuses on weight pruning; 2. HRank evaluates ranks of feature maps, but we evaluate ranks of weight tensors; 3. HRank uses feature rank as filter saliency; our work uses weight rank to guide the update of a sparse network topology.

# 3 Experiments

Our Rank-based PruninG (RPG) method is evaluated on several behchmarks and proved outstanding among recent unstructured pruning baselines. This section presents the experiment results to empirically prove the effectiveness of our RPG method, especially on high sparsities. First, we will show the results of RPG on two image classification datasets: the comparatively small-scaled CIFAR-10, and the large-scaled ImageNet. Then, we will present the results of RPG on downstream vision tasks. Finally, an ablation study will be given.

## 3.1 CIFAR Experiments

**Experiment settings.** We first compare our RPG pruning method with other methods on CIFAR-10 classification. CIFAR-10 is one of the most widely used benchmark for image classification. It consists of 60000 $32 \times 32$ images: 50000 for training, and 10000 for validation. We hope to try our RPG method first on this relatively small dataset and look for heuristic patterns.

Among the pruning baselines, we choose ProbMask [59] and AC/DC [40]for comparison because these two methods are intended for high-sparsity pruning. Additionally, ProbMask is a recent baseline that provides both CIFAR and ImageNet classification results, enabling us for further investigation on larger-scale datasets. Other baselines including PBW [19] and MLPrune [55] are earlier conventional pruning baselines for references. For fair comparison, our RPG method is applied to modern CNN structures, *i.e.* VGG-19 and ResNet-32, and prune for 300 epochs, according to the setting of ProbMask [59]. The results are shown in Table 1.

**Results analysis.** At relatively low sparsities, the gap between recent baselines are small. ProbMask [59], AC/DC [40], and RPG all give satisfactory results at 99% compared with early pruning works. But as sparsity further increases, the three methods undergo significant performance decay on

either network. At 99.5% and 99.9%, our RPG method shows great advantage over the other two baselines. This discovery inspires us further investigate the high-sparsity potential of RPG on the large-scale ImageNet dataset.

## 3.2 ResNet-50 on ImageNet

**Experiment settings.** Sparse ResNet-50 networks evaluated on the ImageNet dataset are the most commonly-used and recognized weight pruning benchmarks. ImageNet ISLVRC2012 [9] is a large scale image classification dataset. It contains 1281K images in the training set and 50K images in the validation set. All the images are shaped $224 \times 224$ and distributed in 1000 classes. ResNet-50 [22] is a medium-size canonical CNN with 25.5M parameters and 8.2G FLOPs, designed for ImageNet classification.

Our RPG method is applied on ResNet-50 under high sparsities: 80%, 90%, 95%, and 98%. We compare RPG with recent baselines. Among the baselines, STR [29] automatically learns pruning sparsity; WoodFisher [48], GraNet [34] and ProbMask [60] are methods based on gradual pruning; AC/DC [40] and ProbMask [60] are baselines targeted at high sparsities; PowerPropagation [45] is an improvement of Top-KAST [26] that relies on a pre-set layerwise sparsity distribution. For fair comparison, all results are 100-epoch baselines; we used standard ImageNet configs, detailed in the Appendix. The results are presented in Table 2. The advantage of adversarial rank-based pruning is manifested at high sparsities.

**Results analysis.** Our method could achieve outstanding performance for sparsities 90%, 95%, and 98%. At lower sparsities (*e.g.* 80%, 90%), Wood-Fisher [48] takes the lead among the baselines. Our RPG method is slightly lower than WoodFisher [48] by 0.07% in ImageNet accuracy at 80% sparsity. At higher sparsities, our method outcompetes other baselines. Other competitive baselines at high sparsities include PowerPropagation [45] and AC/DC [40]. However, the gap between our RPG method and these baselines widened at high sparsities. Specificlly, our method outperforms current top baseline by 1.13% of ImageNet Top-1 accuracy at 98% sparsity.

| Algorithm | Sparsity | Accuracy |
|---|---|---|
| ResNet-50 [22] | 0 | 76.80 |
| STR [29] | 0.8 | 76.19 |
| WoodFisher [48] | 0.8 | **76.73** |
| GraNet [34] | 0.8 | 76.00 |
| AC/DC [40] | 0.8 | 76.30 |
| PowerPropagation [45] | 0.8 | 76.24 |
| RPG (Ours) | 0.8 | 76.66 |
| STR [29] | 0.9 | 74.31 |
| WoodFisher [48] | 0.9 | 75.26 |
| GraNet [34] | 0.9 | 74.50 |
| AC/DC [40] | 0.9 | 75.03 |
| ProbMask [59] | 0.9 | 74.68 |
| PowerPropagation [45] | 0.9 | 75.23 |
| RPG (Ours) | 0.9 | **75.80** |
| STR [29] | 0.95 | 70.40 |
| WoodFisher [48] | 0.95 | 72.16 |
| AC/DC [40] | 0.95 | 73.14 |
| ProbMask [59] | 0.95 | 71.50 |
| PowerPropagation [45] | 0.95 | 73.25 |
| RPG (Ours) | 0.95 | **74.05** |
| STR [29] | 0.98 | 62.84 |
| WoodFisher [48] | 0.98 | 65.55 |
| AC/DC [40] | 0.98 | 68.44 |
| ProbMask [59] | 0.98 | 66.83 |
| PowerPropagation [45] | 0.98 | 68.00 |
| RPG (Ours) | 0.98 | **69.57** |

Table 2. Sparsified ResNet-50 on ImageNet. All results are official reports from the original works. Best and second best results are **bolded** and underlined.

Erdos-Renyi-Kernel (ERK) [14] is a layerwise sparsity distribution that is commonly used for performance boosting in weight pruning methods that require a pre-set sparsity distribution. However, ERK-based sparse models are computationally costly. Differently, RPG automatically maintains a more balanced sparsity throughout the whole network under the same total sparsity constraint. Though our sparse model slightly lags behind the current ERK variant of SOTA [45] under lower sparsities in certain accuracy, it is much cost-effective. Quantatitively, for 80% sparse ResNet-50, the reported ERK-based State-of-the-Art ImageNet accuracy is merely 0.10% higher than our RPG method (reaching 76.76% for [45]), but costing an extra 58% of FLOPs. The advantage of our RPG method over ERK-based methods is clearly illustrated in Figure 3, where we compare RPG with the ERK variant of TOP-KAST [26] and the State-of-the-Art PowerPropagation [45].

DeepSparse [28] is a recent sparse acceration framework on CPU that makes unstructured-sparse network accerlation possible in applications. We time sparse ResNet-50 on DeepSparse for single-image inference. Results in Table 3 shows that highly-sparse ResNet-50 could achieve around

| Sp. | Acc. | Runtime |
|---|---|---|
| Dense | 76.80 | 40.25ms |
| 0.8 | 76.66 | 39.26ms |
| 0.9 | 75.80 | 27.98ms |
| 0.95 | 74.05 | 22.20ms |
| 0.98 | 69.57 | 20.89ms |

Table 3. Sparse acceleration of sparse ResNet-50 on DeepSparse. Unstructured pruning could bring **2×** acceleration effects on CPU at high sparsities.

| Algorithm | Sp. | BoxAP | MaskAP |
|---|---|---|---|
| Mask R-CNN | 0 | 38.6 | 35.2 |
| RigL [14] | 0.5 | 36.4 | 32.8 |
| AC/DC [40] | 0.5 | **37.9** | **34.6** |
| RPG (Ours) | 0.5 | 37.7 | 34.4 |
| RigL [14] | 0.7 | 32.3 | 29.1 |
| AC/DC [40] | 0.7 | 36.6 | 33.5 |
| RPG (Ours) | 0.7 | **37.6** | **34.4** |
| RigL [14] | 0.8 | 26.0 | 23.7 |
| AC/DC [40] | 0.8 | 34.9 | 32.1 |
| RPG (Ours) | 0.8 | **37.1** | **33.8** |

Table 4. Mask R-CNN pruning on COCO val2017. "Sp." stands for model sparsity. Best results are **bolded**.

| Algorithm | Sp. | Accuracy |
|---|---|---|
| DeiT-S [52] | 0 | 79.85 |
| SViT-E [6] | 0.5 | 79.72 |
| AC/DC [40] | 0.5 | **80.15** |
| RPG (Ours) | 0.5 | **80.15** |
| SViT-E [6] | 0.6 | 79.41 |
| AC/DC [40] | 0.6 | 79.69 |
| RPG (Ours) | 0.6 | **79.89** |
| AC/DC [40] | 0.8 | 76.24 |
| RPG (Ours) | 0.8 | **77.42** |

Table 5. Sparse DeiT-S on ImageNet. "Sp." stands for model sparsity. The best results are **bolded**.

$2\times$ accerlation on CPU. This observation reveals that highly unstructured-sparse networks have promising applicative prospects on edge devices that could not afford power and cost-intensive GPUs, e.g. micro robots, wearable devices, et cetera. These devices feature limited memory and power, but high inference speed demands. In this sense, our RPG unstructured pruning method is of great application value.

## 3.3 Downstream Vision Tasks

We also test our weight pruning method on downstream vision tasks. Mask R-CNN [21] is a widely used benchmark for conventional downstream tasks, namely, object detection and instance segmentation. We try to apply our weight pruning method to Mask R-CNN and compare its detection and segmentation performance against other pruning baselines. As for the choice of baselines, we found that limited weight pruning works conducted experiments on downstream vision tasks. We choose the following baselines for comparison: RigL [14] is a commonly used sparse-to-sparse baseline. AC/DC [40] is good at high-sparsity pruning on ImageNet classification. All methods are applied on Mask R-CNN ResNet-50 FPN variants to measure the mAP for bounding boxes and segmentation masks.

For all Mask R-CNN experiments, we follow the official training of COCO $1\times$ [21]: pruning and finetuning lasts for 90K iterations in total. The pruning results evaluated on COCO val2017

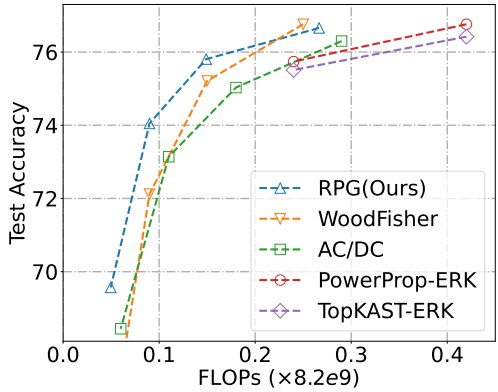

Figure 3. ImageNet accuracy versus FLOPs on sparse ResNet-50. Our method achieves better Accuracy-FLOPs trade-off compared with competitive pruning baselines, especially at high sparsities.

are illustrated in Table 4. Similar to the trend in classification experiments, our RPG method gains an advantage at high sparsities compared with AC/DC [40]. As sparsity increases from $70\%$ to $80\%$, the gap between AC/DC and RPG widens from $1.0$ to nearly $2.0$ for both detection and segmentation mAPs. This finding shows that RPG is a weight pruning method that could be generalized to various vision tasks: it always works well at high sparsities without the need for significant modifications.

## 3.4 Vision Transformers

Recent works on vision model architectures focus on transformers [13, 52]. Transformer architecture models are proven particularly effective on large-scale image recognition tasks and are well applied

to various downstream tasks [4, 58, 5], but they are still struggling for industrial applications due to large model size and computation cost. To address these problems, works like SViT-E [6] attempted to apply unstructured pruning on vision transformers.

Though our method is not specifically designed for models with the attention mechanism, we explore the effect of our weight pruning method on DeiT-S [52] and compare it with high-sparsity weight pruning baseline [40] and the transformer pruning baseline [6] in Table 5. For fair comparison, all pruning experiments follow the setting of SViT-E [6]: the DeiT-S model is pruned for 600 epochs on ImageNet [9]. All other settings are identical the official training setting of [52], including batchsize, learning rate, etc.

### 3.5 Ablations

In this section, we inspect the effect of rank loss. The rank-based pruning objective involves an affine parameter $\lambda$ that controls the amount of rank loss with respect to the original task loss. When $\lambda = 0$, rank loss is turned off. Investigating the relations of rank versus $\lambda$ and accuracy versus $\lambda$ on a ResNet-32 of 99.5% sparsity as shown in Figure 4, we found rank loss could significantly increase the average rank throughout all layers of the sparse network. A substantial increase of accuracy is also observed. But as $\lambda$ further increases, the average rank will be saturated. Reversely, as $\lambda$ further increases, the classification accuracy will decrease. This could be attributed to the property of affine combination in Equation (2.7). When $\lambda$ is large, the pruning objective will pay too much attention to maintain weight ranks and neglect the goal of performing the task well. Hence, it is necessary to tune $\lambda$ and find the most optimal one.

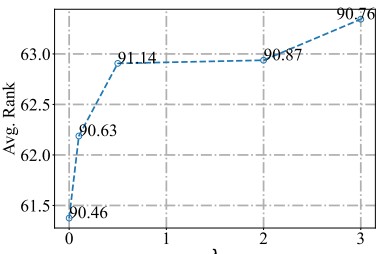

Figure 4. Average weight matrix rank of ResNet-32 [22] versus affine hyperparameter $\lambda$. Accuracies on CIFAR-10 are marked.

| Type | Time | FLOPs |
|------|------|-------|
| SVD | 16.5min | 5.07e15 |
| RPG90% | 1003min | 1.34e18 |

Table 6. SVD overhead compared with the overall pruning & finetuning cost of RPG on 90% sparse ResNet-50.

| Baseline | Sparsity | Train FLOPs |
|----------|----------|-------------|
| ResNet-50 | (Dense) | 3.14e18 |
| AC/DC | 0.9 | 0.58× |
| PowerProp. | 0.9 | 0.49× |
| RPG(Ours) | 0.9 | **0.43×** |

Table 7. Training FLOPs comparison with pruning baselines on sparse ResNet-50.

### 3.6 Overhead Analysis

As introduced in Section 2.4, RPG involves costly SVD calculations. However, we conduct experiments and illustrate that SVD accounts for very minimal cost overhead during pruning in terms of both time and FLOPs. As shown in Table 6, the overall time and FLOPs for SVD calculations only accounts for $< 2\%$ of the whole RPG pruning cost. We also compare the FLOPs overhead of RPG with other pruning methods. Observing from Table 7, our method is the most cost-effective compared with baselines. Above all, the extra overhead brought by rank loss calculations is not a concern.

## 4 Conclusion

This paper proposes the Rank-based Pruning (RPG) method. We investigate weight pruning from a matrix rank perspective, and yield the observation that higher ranks of sparse weight tensors could yield better performance. Based on this heuristic, the adversarial rank loss is designed as the optimization objective that guides the mask update process during pruning. In this manner, our method prunes weight tensors in a rank-favorable fashion. Our RPG method is experimented on various settings and outperforms various baselines.

## 5   Limitations

Unstructured pruning has limited acceleration effect on GPU devices. New GPU architectures or GPU sparse acceleration supports are needed to exert the speed potential of our unstructured pruning method on GPUs.

**Acknowledgement**

This work is supported by National Key R&D Program of China under Grant No.2022ZD0160304. We gratefully acknowledge the support of MindSpore [38], CANN (Compute Architecture for Neural Networks) and Ascend AI Processor used for this research. We also gratefully thank Yuxin Zhang and Xiaolong Ma for their generous help.

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

# Appendix

## A Proofs of Theorems

### A.1 Proof of Theorem 1

**Theorem 3** (The best low-rank approximation). *Suppose $W$ is decomposed via SVD and yield $W = \sum_{i=1}^{r} \sigma_i u_i v_i^T$ where singular values $\{\sigma_i\}$ are sorted in descending order. Given integer $k < r$, the best $k$-rank approximation of $W$, namely the $k$-rank matrix that has the smallest $l_2$ distance to $W$ is*

$$\widetilde{W} = \sum_{i=1}^{k} \sigma_i u_i v_i^T.$$

*This theorem is also the Eckart-Young-Mirsky Theorem for Frobenius norm.*

Though the proof is easily accessible (*e.g.* from Wikipedia [2]), we will provide a sketch of proof here for reference.

*Proof.* Denote $W_k = \sum_{i=1}^{k} \sigma_i u_i v_i^T$ be the best $k$-rank approximation of $W$, $\sigma_i(W)$ be the $i^{th}$ largest singular value of $W$. Then the low-rank approximation error could be reduced as follows:

$$\left\| W - \widetilde{W} \right\|_F^2 = \left\| \sum_{i=k+1}^{r} \sigma_i u_i v_i^T \right\|_F^2 = \sum_{i=k+1}^{r} \sigma_i^2. \tag{A.1}$$

Given $W = W' + W''$, according to the triangle inequality of spectral norm,

$$\sigma_1(W) = \sigma_1(W') + \sigma_1(W''). \tag{A.2}$$

Then for two arbitrary ranks $i, j \geq 1$, we have:

$$\begin{aligned}
\sigma_i\left(W'\right) + \sigma_j\left(W''\right) &= \sigma_1\left(W' - W'_{i-1}\right) + \sigma_1\left(W'' - W''_{j-1}\right) \\
&\geq \sigma_1\left(W - W'_{i-1} - W''_{j-1}\right) \\
&\geq \sigma_1\left(W - W_{i+j-2}\right) \\
&= \sigma_{i+j-1}(W).
\end{aligned} \tag{A.3}$$

Assume there is another $k$-rank approximation $X$, Then according to the above formula, for arbitrary $i \geq 1$,

$$\sigma_i(W - X) = \sigma_i(W - X) + \sigma_{k+1}(X) \geq \sigma_{k+i}(W) \tag{A.4}$$

Hence,

$$\|W - X\|_F^2 \geq \sum_{i=1}^{n} \sigma_i\left(W - X\right)^2 \geq \sum_{i=k+1}^{n} \sigma_i^2, \tag{A.5}$$

which means $\widetilde{W}$ is the best $k$-rank approximation. $\qquad\square$

### A.2 Proof of Theorem 2

Theorem 2 states the effectiveness of rank loss. Before the theorem, recall notations that we previously defined: $\overline{W} := \frac{W}{\|W\|}$ is the $l_2$ normalized weight matrix $W$; $U, \Sigma, V$ are matrices reached from the SVD of $\overline{W}$, where $U = \{u_1, u_2, ...\}$ and $V = \{v_1, v_2, ...\}$ are orthonormal bases; $\Sigma$ is a diagonal matrix where singular values $\{\sigma_1, \sigma_2, ...\}$ are sorted in descending order on the diagonal; operator $\text{Trun}\left(U\Sigma V^T\right) = \sum_{i=1}^{k} \sigma_i u_i v_i^T$ stands for $k$-rank truncated SVD, or the $k$-rank best approximation of $\overline{W}$ according to Theorem 1.

---

[2]https://en.wikipedia.org/wiki/Low-rank_approximation

**Theorem 4** (Effectiveness of the adversarial rank loss). *Given the adversarial rank loss*

$$\mathcal{L}_{rank} = -\|\overline{W} - \text{Trun}\left(U\Sigma V^T\right)\|_F^2. \tag{A.6}$$

*If we optimize $W$ in rank loss via gradient descent, the rank of $W$ will increase.*

*Proof.* In gradient descent, the update from weight $W$ to $W'$ based on rank loss $\mathcal{L}_{rank}$ could be described as:

$$W' = W - \gamma \frac{\partial \mathcal{L}_{rank}}{\partial W}. \tag{A.7}$$

We first simplify rank loss. Since $U = \{u_1, u_2, ...\}$ and $V = \{v_1, v_2, ...\}$ are orthonormal bases, we could easily rewrite rank loss $\mathcal{L}_{rank}$ as the squared sum of small singular values:

$$
\begin{aligned}
\mathcal{L}_{rank} &= -\|\frac{W}{\|W\|_F} - \sum_{i=1}^{k} \sigma_i u_i v_i^T\|_F^2 \\
&= -\|U\Sigma V^T - U\Sigma_{[1:k]}V^T\|_F^2 \\
&= -\|\Sigma - \Sigma_{[1:k]}\|_F^2 = - \sum_{i=k+1}^{r} \sigma_i^2.
\end{aligned} \tag{A.8}
$$

The form Equation (A.8) allows us to apply chain rule to calculate the gradient of the normalized weight matrix $\frac{\partial \mathcal{L}_{rank}}{\partial \overline{W}}$:

$$\frac{\partial \mathcal{L}_{rank}}{\partial \overline{W}} = \sum_{i=k+1}^{r} \frac{\partial \mathcal{L}_{rank}}{\partial \sigma_i} \frac{\partial \sigma_i}{\partial \overline{W}} = - \sum_{i=k+1}^{r} 2\sigma_i u_i v_i^T. \tag{A.9}$$

Again, the chain rule is applied for the derivative of the weight matrix. For clarity, we show the gradient expression for a single weight parameter $W[m, n]$ (the weight value at position $(m, n)$ in the reshaped weight matrix $W$):

$$
\begin{aligned}
\frac{\partial \mathcal{L}_{rank}}{\partial W[m, n]} &= tr\left(\frac{\partial \mathcal{L}_{rank}}{\partial \overline{W}^T} \frac{\partial \overline{W}}{\partial W[m, n]}\right) \\
&= -\frac{(\sum_{i=k+1}^{r} 2\sigma_i u_i v_i^T)[m, n]}{\|W\|_F} \\
&\quad + \frac{W[m, n] \sum \left(W \odot \left(\sum_{i=k+1}^{r} 2\sigma_i u_i v_i^T\right)\right)}{\|W\|_F^3}.
\end{aligned} \tag{A.10}
$$

Based on the gradient, one step of optimization under learning rate $\alpha$ could be expressed in a neat matrix multiplication format, decomposed by orthonormal bases $U = \{u_1, u_2, ...\}$ and $V = \{v_1, v_2, ...\}$.

**Lemma 1** (Weight optimization based on rank loss). *Denote $\sum \left(W \odot \left(\sum_{i=k+1}^{r} 2\sigma_i u_i v_i^T\right)\right) := c$, which is a scalar constant within each step, one step of weight $W$ optimization under learning rate*

$\gamma > 0$ *and rank loss (as defined in Equation (2.6)) could be expressed as:*

$$
\begin{aligned}
W' &= W - \gamma \frac{\partial \mathcal{L}_{rank}}{\partial W} \\
&= W - \gamma \left( -\frac{\sum_{i=k+1}^{r} 2\sigma_i u_i v_i^T}{\|W\|_F} + \frac{cW}{\|W\|_F^3} \right) \\
&= U\Sigma V^T + \frac{2\gamma}{\|W\|_F} U\Sigma_{[k+1:r]} V^T - \frac{c\gamma}{\|W\|_F^3} U\Sigma V^T \\
&= U \left( \left(1 - \frac{c\gamma}{\|W\|_F^3}\right)\Sigma + \frac{2\gamma}{\|W\|_F}\Sigma_{[k+1:r]} \right) V^T.
\end{aligned}
\tag{A.11}
$$

From the formula of the optimized weight $W'$, we can reach the following conclusions on the optimized weight $W'$: firstly, $W'$ promises the same set of orthonormal bases $U, V$ after SVD; secondly, comparing small singular values against others, all singular values are penalized by the same amount $\frac{c\gamma}{\|W\|_F^3}$; but the small singular values (ranking from $k+1$ to $r$) are awarded with increments $\frac{2\gamma}{\|W\|_F}$. Regardless of swapped singular values due to magnitude change (because of small learning rate $\gamma$), small singular values will make up more proportion in all the singular values after one step of update. Recall the definition for $\delta$-rank, given fixed sum of squared singular values after $l_2$ normalization, the rank of $W$ will increase. $\qquad\square$

# B  Method Details

## B.1  Details on Gradient-Grow.

We explain in details about the procedure of gradient grow evolved from RigL [14]. At each gradient grow step, first we calculate the Rank-based Pruning (RPG) objective $\mathcal{L}$. Then we back-propagate $\mathcal{L}$ for gradients of network weights. Finally, gradients of pruned network weights are sorted in magnitudes; weights with large gradients are re-activated.

The number of weights to be re-activated are determined by the number of remaining weights. The whole pruning framework is detailed in Algorithm 1. Grow fraction $\alpha$ is a function of training iterations that gradually decays for stability of training. Cosine Annealing is used for $\alpha$, following [14].

## B.2  Selection of Approximation Rank

Factor $k$ of the truncation operator $\mathrm{Trun}$ controls the rank of the low-rank approximation in this adversarial process. However, controlling $k$ for each layer is hardly practical, because layers are large in quantity and vary in shapes. We leverage the concept of $\delta$-rank in Definition 1, and tend to control the approximation error (also the negative rank loss $-\mathcal{L}_{rank}$ for a layer) rather than control $k$. We set a constant $\tilde{\delta}$ between $(0, 1)$ as the target approximation error between the normalized weight matrix $\overline{W}$ and its low rank counterpart. Then we find the best $k$ that has the closest low-rank approximation error as $\tilde{\delta}$ for each layer. Mathematically, this selection process could be written as:

$$
\arg\min_k | - \mathcal{L}_{rank} - \tilde{\delta}|.
\tag{B.1}
$$

# C  Experiment Details

## C.1  CIFAR Experiments

In the CIFAR experiments, we train VGG-19 and ResNet-32 for 300 epoch, which is identical to Zhou et al. [59]. We use the SGD optimizer with momentum 0.9, batchsize 128, learning rate 0.1, and weight decay 0.005.

**Algorithm 1:** Rank-based Pruning (RPG)

---

**Input** : A dense model $W$ with $n$ layers; Target density (a function of iteration) $d$; Grow
fraction (a function of iteration) $\alpha$; Mask update interval $\Delta T$; Total training iteration
$T = T_{prune} + T_{finetune}$

**Output :** A Sparse Model $W \odot M$

---

1 // Initialize a dense mask ;
2 $M \leftarrow \mathbb{1}$;
3 // Stage 1: prune and grow;
4 **for** $t \leftarrow 1$ *to* $T_{prune}$ **do**
5     Forward propagation for minibatch loss $\mathcal{L}_{task}$;
6     **if** $t\%\Delta T = 0$ **then**
7         // Update mask $M$;
8         Calculate and sum layerwise rank loss;
9         Keep top $d_t$ proportion of weights in $W$ globally to get layerwise density $d_t^i$;
10         **for** $i \leftarrow 1$ *to* $n$ **do**
11             Prune to density $(1 - \alpha_t)d_t^i$ based on $|W|$;
12             Grow to density $d_t^i$ again based on $|\nabla \mathcal{L}|$;
13         **end for**
14     **end if**
15     **else**
16         Train sparse net with task loss;
17     **end if**
18 **end for**
19 // Stage 2: keep masks fixed and finetune;
20 **for** $t \leftarrow T_{prune} + 1$ *to* $T$ **do**
21     Keep mask $M$ static, and train weight $W$ till converge;
22 **end for**
23 **return** *Weights of the Pruned Model* $W \odot M$

---

## C.2   ResNet-50 on ImageNet

We mainly follow Goyal et al. [18] for ImageNet experiments. ImageNet experiments are run on 8 NVIDIA Tesla V100s. Our RPG method is applied on ResNet-50 for 100 epoch, and compared with 100-epoch baselines at various sparsities. We use the SGD optimizer with momentum 0.9, 1024 batchsize, a learning rate of 0.4, 0.0001 weight decay, 0.1 label smoothing, 5 epochs of learning rate warmup. We choose 1024 batchsize instead of the commonly-used 4096 batchsize [14, 45] due to GPU memory constraints. We did not use augmentation tricks like mixup or cutmix. Standard pretrained model from torchvision is used in ImageNet setting for fair comparison with top baselines. $\alpha$ is kept at 0.3; $\Delta T$ is set to 100; $T_{prune}$ is set to 90 epochs.

## C.3   Downstream Vision Tasks

In Mask R-CNN experiments, all methods are implemented by us on the identical settings for fair comparison. We follow the training setting of the original Mask R-CNN [21] on Detectron-2 [54]. The three methods are applied for 96000 iterations. Notably, certain components in Mask R-CNN ResNet-50 FPN are loaded from pretrained models and left untrained. These components are omitted from pruning and sparsity calculations for RPG and baseline experiments.

## C.4   Vision Transformers

In vision transformers, we mainly follow the official setting of DeiT [52], but we extend training epochs from 300 to 600 for fair comparison with SViT-E [6]. All other training settings are identical to the official training setting of DeiT [52].

## C.5 Replication of AC/DC

AC/DC [40] achieves State-of-the-Art performance at high sparsities. Therefore, we hope to compare our method extensively with AC/DC on various settings. Accordingly, the schedule of AC/DC need slight modifications based on the original setting. According to the original work [40], dense and sparse status are alternated every 5 epochs. We scale the period according to the ratio of the training epochs versus the standard training epochs in the paper [40]. For example, if the AC/DC method is applied for 300 epochs, since the original ImageNet AC/DC setting has only 100 epochs, we lengthen the period of AC/DC from the original 5 epochs to 15 epochs. For warming-up and finetuning periods, the similar scaling is performed.

## D  Additional Experiments

We also tried RPG pruning without loading pretrained models on ResNet-50 ImageNet experiments. Training settings are kept the same with other ResNet-50 pruning experiments. The results are provided in Table 8. The proposed RPG method could surpass all baselines even without loading pretrained models.

| Algorithm | Sparsity | FLOPs | Accuracy |
|---|---|---|---|
| ResNet-50 [22] | 0 | $1\times$ | 76.80 |
| RPG from scratch | 0.9 | $0.155\times$ | 75.35 |
| RPG from scratch | 0.95 | $0.093\times$ | 73.62 |

Table 8. RPG-Sparsified ResNet-50 from scratch on ImageNet.

