# OpenReview forum: "Towards Higher Ranks via Adversarial Weight Pruning"
_NeurIPS.cc/2023/Conference — NeurIPS 2023 poster_

### Official Review · Reviewer_GHQR · 2023-07-05

**Soundness:** 4 excellent
**Presentation:** 3 good
**Contribution:** 3 good
**Rating:** 7
**Confidence:** 5

**Summary:**

This paper proposes a novel Rank-based PruninG (RPG) method for network pruning that maintains the ranks of sparse weights in an adversarial manner, leading to high-rank topology and improved performance. The proposed method is evaluated on various datasets and tasks, including image classification, object detection, and semantic segmentation, and compared to state-of-the-art pruning methods. The experimental results show that the proposed RPG method outperforms the existing methods in terms of accuracy and efficiency. The paper also provides insights into the importance of rank preservation in network pruning and the potential of adversarial training for improving the performance of pruned networks.

**Strengths:**

+ This paper is well-written and organized, with clear explanations of the proposed method and experimental results. The authors provide detailed insights into the importance of rank preservation in network pruning and the potential of adversarial training for improving the performance of pruned networks.

+ This paper proposes a novel Rank-based PruninG (RPG) method that maintains the ranks of sparse weights in an adversarial manner, leading to high-rank topology and improved performance. This approach is original and creative, as it combines the ideas of rank preservation and adversarial training to address the limitations of existing pruning methods.

+ This paper provides a comprehensive evaluation of the proposed RPG method on various datasets and tasks, including image classification, object detection, and semantic segmentation. The experimental results show that the proposed method outperforms the existing methods in terms of accuracy and efficiency, which demonstrates the quality and effectiveness of the proposed approach.

Overall, this paper has a high-quality and significant contribution to the field of network pruning, with original ideas, rigorous evaluation, clear explanations, and practical implications.


**Weaknesses:**

- One potential weakness of the proposed Rank-based Pruning (RPG) method is that it may only achieve better performance than existing methods when the sparsity rate is high. As mentioned in the paper, the RPG method outperforms existing methods such as WoodFisher, PowerPropagation, and AC/DC at sparsity rates of 90%, 95%, and 98%. However, at lower sparsity rates such as 80%, the RPG method is slightly lower than WoodFisher in terms of ImageNet accuracy. This suggests that the RPG method may not be as effective at lower sparsity rates, and may require higher sparsity rates to achieve better performance than existing methods. This could be a limitation for some applications where lower sparsity rates are preferred due to memory or computational constraints.

- The proposed method can only be applied on weight pruning, which is slower than filter pruning when their pruning rates are same.

**Questions:**

This paper mentions that the proposed method is based on the rank of the weights, but it would be helpful to have more details on how exactly the rank is determined and how it affects the pruning process. Additionally, are there any limitations or potential drawbacks to using rank-based pruning compared to other pruning methods that use different criteria for selecting weights to prune?

**Limitations:**

Yes.

---

> ### Author Rebuttal · Authors · 2023-08-08
>
> Dear Reviewer GHQR,
>
> Thank you very much for your review. Here are our reponses to the weaknesses and questions you raised:
> 1. *The RPG method is not performant at low sparsities, impairing low-sparsity applications*: We admit the limitation of the RPG pruning methods under low-sparsity regimes, because the rank-collapse effect is not manifested on low-sparsity networks. But we hold that highly-sparse networks are more valuable in terms of application. As shown in the ResNet-50 example (cf. Table 3), intermediate-sparse models could achieve little CPU acceleration effect; while the speedup is significant for models at high sparsities. Hence, we could find a better accuracy-speed trade-off on highly-sparse networks, and the application value of the proposed RPG method is demonstrated.
> 2. *About how the rank is determined*: Due to page limits, we put details of rank determination in appendix B.2. Considering weights of different layers vary significantly in shapes, we designed an automatic selection mechanism for ranks.
> 3. *Any limitations or potential drawbacks to using rank-based pruning*: Our method is targeted at the "rank collapse" effect that only occurs on highly-sparse networks. It has limited effect on models with relatively low sparsity, because the "rank collapse" effect is not so serve on low sparsity networks.
>
>
> Sincerely,
>
> Authors

---

### Official Review · Reviewer_P9XJ · 2023-07-06

**Soundness:** 4 excellent
**Presentation:** 3 good
**Contribution:** 3 good
**Rating:** 6
**Confidence:** 5

**Summary:**

This work propose a novel objective for performing element-wise pruning of DNN model. The work identifies the loss of weight rank as the key factor influencing the performance of DNN when pruned to high sparsity. This issue is tackled by including a rank loss in the pruning criteria, so that weight elements contributing to the weight matrix rank are preserved. Experiment results show the proposed method outperform previous work under high sparsity.

**Strengths:**

1. The motivation of preserving high rank in element-wise pruning is novel, and it motivates the proposed method well
2. The proposed method is well formulated and is technically sound
3. Adequate ablation study and extensive experiments are conducted, showing promising results

**Weaknesses:**

One major concern of this paper is whether aiming for higher element-wise sparsity is useful for the deployment of efficient DNN model. As mentioned in the limitation, element-wise sparsity is not well supported on GPU. Even on CPU, as shown in Tab.3 only 2x speedup can be achieved with 95% sparsity, at the cost of 3% accuracy drop. While a 50% structural sparsity can lead to 4x FLOPs reduction, potentially 3x speedup, with less accuracy drop. This would indicate that aiming towards a high-rank element-wise sparsity is not as useful as directly having structural sparsity. I would suggest the author to provide a comparison of accuracy-speed tradeoff of the proposed method and previous structural pruning results, to show the importance of the proposed method.

Minor issue: It would be better to have a pseudo code for the pruning procedure performed in Sec. 2.5

**Questions:**

See weakness

**Limitations:**

The limitation is adequately discussed. No potential negative social impact is observed.

---

> ### Author Rebuttal · Authors · 2023-08-09
>
> Dear reviewer P9XJ,
>
> Thank you very much for your review. Here are our responses:
>
> Q1. *The author should provide a comparison of accuracy-speed tradeoff of the proposed method and previous structural pruning results.*
>
> A1: Thanks for your suggestion. Here we attach a table comparing the CPU speedup of one of the latest structural baseline TPP [Wang et al.] with RPG. The table shows that the gap between the two methods are narrow.
>
> | Baseline | Accuracy  | Speed |
> | --------- | ----- | ----- |
> | RPG | 76.58 | 1.08x  |
> | TPP | 76.44 | 1.13x  |
> | RPG | 75.63 | 1.56x  |
> | TPP | 75.60 | 1.45x  |
> | TPP | 74.51 | 1.86x  |
> | RPG | 73.89 | 1.99x  |
>
>
> Besides, while we admit structural baselines could achieve good accerlation effects on GPU (while unstructured sparsity has little effect), unstructured-sparse networks are more compact in terms of size. The size advantage of unstructured sparsity allows applications in storage or bandwidth-limited scenarios.
>
> Q2. *It would be better to have a pseudo code for the pruning procedure performed in Sec. 2.5.*
>
> A2: Thanks for your advice. The pseudo code is included in the Appendix B.1, pp. 4 due to page limits. We will consider moving it to a prominent place in later revision.
>
> Sincerely,
>
> Authors
>
> References:
>
> [Wang et al.] Trainability Preserving Neural Pruning. ICLR 2023.

---

> ### Comment · Reviewer_P9XJ · 2023-08-17
> **Thanks for the response**
>
> I would like to thank the authors for the response. I would suggest the author to include this result in the revision as a discussion on the limitation of the proposed method. Meanwhile I agree with the author's motivation that unstructured sparsity is still useful in memory-bounded scenario, and the proposed method serves as a good approach to achieve extreme sparsity.
>
> One suggestion I would have for the author is to include additional experiments on performing RPG on top of a structurally-sparsed model. As structural pruning already (presumably) removed unnecessary structures, RGP will be much more effective as it can preserve the remaining important structure. I assume RPG should outperform other unstructural pruning method when applied on already-compressed models.

---

> > ### Author Response · Authors · 2023-08-21
> > **Additional Experiments on Structually-Sparse Model Pruning**
> >
> > Thank you very much for your suggestions, and we tried to conduct experiments as suggested despite limited time left for discussion. We conducted 90\%-sparse pruning experiments on TPP [Wang et al.] structurally-sparse ResNet-50 models, and we compared our RPG pruning method with AC/DC [Peste et al.], a competitive pruning baseline. Experimental settings are kept the same to our ImageNet experiments, and the results are shown in the table below. RPG could outperform AC/DC on a structurally-pruned model.
> >
> > | Methods                      | Accuracy |
> > | ---------------------------- | -------- |
> > | TPP [Wang et al.] (Baseline) | 74.51    |
> > | AC/DC [Peste et al.]         | 71.33    |
> > | RPG (Ours)                   | **71.77**    |
> >
> > [Wang et al.] Trainability Preserving Neural Pruning. ICLR 2023.
> >
> > [Peste et al.] AC/DC: alternating compressed/decompressed training of deep neural networks. NeurIPS 2021.

---

### Official Review · Reviewer_jsvG · 2023-07-06

**Soundness:** 3 good
**Presentation:** 3 good
**Contribution:** 3 good
**Rating:** 7
**Confidence:** 4

**Summary:**

This paper proposes a new weight pruning method for compressing Convolutional Neural Networks (CNNs) called Rank-based PruninG (RPG). The RPG method consists of two steps: first, the low-rank approximation error for the weight matrices is minimized using singular value decomposition, and second, the weight matrices are pushed away from their low-rank approximation to maximize their distance. The authors demonstrate that the RPG method outperforms other state-of-the-art methods in terms of accuracy and compression rate on various datasets and tasks.

**Strengths:**

Originality: Pruning weights while maintaining the ranks of sparse weights in an adversarial manner is original and has not been proposed before, which is different from other pruning methods that focus on removing individual weights or neurons.

Quality: The experimental results show that the RPG method achieves higher accuracy and compression rate than the previous baselines. The authors also provide insights into the mechanism of the RPG method and its impact on the network structure, which enhances the quality of the paper.

Clarity: The paper is well-written and easy to understand, which enhances its clarity. The authors provide clear explanations of the proposed method and its implementation. They also provide detailed experimental results and analysis.

Significance: When the pruning rate is high, traditional pruning methods can lead to a structured pattern in the remaining weights, which limits their performance. The proposed rank-based pruning method maintains the ranks of sparse weights in an adversarial manner, which ensures that the pruned network retains its structure and performance even at high pruning rates.

**Weaknesses:**

1) The adversarial optimization involves a min-max problem that requires additional computation and optimization steps, which can increase the training time and complexity. Similarly, SVD is a computationally expensive operation that requires matrix factorization and singular value decomposition, which can also increase the training cost. Moreover, the proposed method requires the computation of low-rank approximations and the search for the best k-rank approximation, which can further increase the training cost. These additional computations and operations can make the proposed method less practical for large-scale networks or real-time applications.


2) Singular value decomposition (SVD) is used in the proposed method to compute the rank and low-rank approximations of weight matrices. In contrast, Canonical Polyadic Decomposition (CPD) and Tucker Decomposition are other methods for decomposing tensors into lower-rank components. It is better to explain why choosing SVD to estimate the rank of weights.

**Questions:**

Please refer to the weaknesses.

**Limitations:**

The authors have adequately addressed the limitations.

---

> ### Author Rebuttal · Authors · 2023-08-09
>
> Dear reviewer jsvG,
>
> Thank you very much for your review. Here are our responses:
>
> Q1. *Additional computations and operations can make the proposed method less practical for large-scale networks or real-time applications.*
>
> A1: In fact, these extra overheads only accounts for a small proportion of training cost. Firstly, the extra cost is amortized because the procedures are carried out once every one hundred iterations; Secondly, weight SVD, which is much more costly than adverserial loss calculation and best k-rank searching, only accounts <<1\% of the whole pruning cost both in terms of time and FLOPs (according to Sec. 3.6). In a nutshell, additional computations are minimal and won't impact the application value of RPG on large models and other applications.
>
>
> Q2. *Why choosing SVD instead of CPD or Tucker Decomposition?*
>
> A2: Unlike Singular Value Decomposition of matrices, Canonical Polyadic Decomposition and Tucker Decomposition are decomposition methods of high-order tensors (in fact, CPD could be viewed as a high-order extension of SVD). Matrix-form (2-dimensional) weights (rather than high-order tensors) are the most general and widely-applied form in all sorts of neural networks. Hence, we adopt SVD as the decomposition method.
>
> Sincerely,
>
> Authors

---

### Official Review · Reviewer_yERC · 2023-07-09

**Soundness:** 3 good
**Presentation:** 3 good
**Contribution:** 3 good
**Rating:** 6
**Confidence:** 3

**Summary:**

This paper proposes a novel unstructured pruning method, trying to maximize the matrix rank while trying to remove as many model weights as possible. The paper first demonstrates the phenomenon that unstructured pruning may degrades to structured pruning at large sparsity ratios, which is closely related to the fact that the pruned weight matrices become low-rank matrices after many weights are set to zero. Thus, the objective of the proposed method is trying to on the one hand, minimize the task-related loss, and on the other, maximize the rank of the pruned weight matrices, which forms a min-max problem. This min-max problem is then integrated into model pruning via a matrix rank-boosting regularization term. With the gradual pruning framework, the proposed method (RPG) is examined empirically on CIFAR-10, ImageNet, COCO datasets with CNNs and ViTs model architectures. The results show the effectiveness of the RPG.

**Strengths:**

This paper studies the unstructured pruning from the perspective of rank maintenance, which is very novel. The authors made very informative and helpful illustrations to help the readers understand this paper without difficulty. Therefore, the presentation is also very good. Extensive experiments were conducted on different tasks, and the results look good. In summary, this work is a very good attempt to connect model pruning with weight ranks in a novel perspective.

**Weaknesses:**

I listed some weakness of this work from different perspectives. I will consider to raise my score if they are addressed properly.
1. [Motivations] The motivations of this paper by "removing the structuring patterns in the unstructured pruning" do not seem very direct to me. In general, currently the results of the unstructured pruning can not be directly used for hardware acceleration. Therefore, one important research direction is to generate structured mask from the unstructured pruning results, such as [ICML22] Coarsening the Granularity: Towards Structurally Sparse Lottery Tickets. Therefore, I wonder if the motivation of this paper will make the unstructured pruning less adaptable to hardwares?
2. [Presentations] I would suggest the authors to remove some discussions on SVD, which are very basic knowledge in linear algebra. In contrast, more important information in the Appendix can be brought back to the main paper.
3. [Method] It is a bit vague to me, how the gradients are dealt with when Eq. (2.6) is treated as the regularization term. Is $\text{Trunc}(U\Sigma V^T)$ treated as a constant value during back-propagation? Also, I suggest the authors explicitly write down the $\mathcal{L}(\text{task})$ as well, which shows how the sparsity of the model weights are imposed.
4. [Method] In Eq. (2.6), the model weights are denoted by one variable $\mathbf{W}$. However, as we know the model weights may contain many layers and many types (convolutional kernels, fully connected layers, etc.) Do the authors sum them up in implementation using the same coefficient? Or there are different weights assigned to different layers. It is not clear in the paper regarding this point.
5. [Experiments] The baselines used in Tab. 1 and Tab. 2 are not consistent. Are their any specific reasons? The same issue is also spotted in Tab. 5, where in the last row block the method "SViTE" is missing, but the authors stated "For fair comparison, all pruning experiments follow the setting of SViTE." (Line 351).
6. [Experiments] It would help the readers to better compare the results if the authors can include the computational (training) time of different methods.
7. [Minor] There are unnecessary margins under Tab. 2 and Fig. 3. Please consider to remove them and make the layout of the figure better.
8. [Minor] Line "is illustrated in" -> "**are** illustrated in"

**Questions:**

Below is a summary of my comments in the section "Weaknesses".
1. What is the relationship between the rank of the pruned weight matrix? Will the unstructured pruning with higher rank impairs the further acceleration on hardwares?
2. How are the SVD-reconstructed terms dealt with during back-propagation?
3. What are the formulation of $\mathcal{L}(\text{task})$?
4. How are the weights with multi-layers processed to compute Eq. (2.6)?
5. Why are the baselines not consistent within one table/across different tables on the same tasks?
6. What how does the training efficiency of different methods look like?

**Limitations:**

I do not have additional comments on the limitation of this work. Please refer to the "Weaknesses" and "Questions" sections.

---

> ### Author Rebuttal · Authors · 2023-08-09
>
> Dear reviewer yERC,
>
> Thank you very much for your suggestions. Here are the answers to the questions you raised:
>
> Q1. *RPG will make the unstructured pruning less adaptable to hardwares.*
>
> A1: Sorry for ambiguities in the paper. "Structured pattern" does not necessarily mean the acceleration-friendly structural sparsity (which is only a special case of structured pattern); in the context of RPG, "structured pattern" is reflected by the low-rank characteristics of weights. The RPG method mainly focuses on improving weight ranks instead of the actual removal of structural sparsity. We conducted speedtest experiments and compared RPG with the competitive pruning baseline of AC/DC [Peste et al.]. Results show that despite higher ranks, RPG won't affect the hardware adaptability of unstructured-sparse networks.
>
> | ResNet-50 | Acc.  | Speedup | Rank  |
> | --------- | ----- | ------- | ----- |
> | Dense     | 76.80 | 1.00x | 263.5 |
> | AC/DC 95% | 73.14 | 1.66x | 234.2 |
> | RPG 95%   | 73.89 | 1.99x | 262.2 |
>
>
>
> Q2. *How are the SVD-reconstructed terms dealt with during back-propagation?*
>
> A2: The term $Trunc(U\Sigma V)$ is treated as a constant value. The SVD-reconstructed terms are detached during loss calculation.
>
> Q3. *What is the formulation of $\mathcal{L}(task)$?*
>
> A3: $\mathcal{L}(task)$ is exactly the training loss for the original dense network. For instance, $\mathcal{L}(task)$ is cross entropy loss for ResNet-50/DeiT ImageNet classification.
>
> Q4. *How are the weights with multi-layers processed to compute Eq. (2.6)?*
>
> A4: The losses for each layer are summed-up together.
>
> Q5. *Inconsistency of baselines in Tab. 1 and Tab 2; missing SViTE in Tab. 5.*
>
> A5: The inconsistancy between Table 1 and 2 is due to the lack of official reports or code. Sparse ResNet-50 for ImageNet classification is a commonly-used weight pruning benchmark (Table 2), but only a handful of papers report CIFAR results (Table 1). Experiment settings (e.g. models, training epochs) also varies among papers with CIFAR. Hence, we have to follow the setting of one reliable and competitive baseline, namely, ProbMask [Zhou et al.] for CIFAR-10 experiments to guarantee fair comparison. In spite of this, we still tried to include competitive pruning baselines, e.g. we re-implemented AC/DC [Peste et al.] on CIFAR. We attempted to re-implement some other strong baselines but encountered problems (e.g. incomplete open-sourcing; closed-source codebase though requested via emails; failure of replication).
>
> The method SViTE is missing in the last row block, because the SViTE paper [Chen et al.] only reports values on relatively low sparsities (sparsity 50/60). We compared our RPG method to a more powerful baseline AC/DC to demonstrate the strong capability of our method under a high sparsity regime. For 80\% sparsity experiments, we still follow the same setting as SViTE except for the pruning rate.
>
> Q6. *Authors should include the computational (training) time of different methods.*
>
> A6: We provide the running time of some pruning algorithms in the table below:
>
> | Baseline | Acc.  | Time |
> | --------- | ----- | ------- |
> | Dense     | 76.80 | 821 min |
> | AC/DC 90% | 75.03 | 861 min |
> | RPG 90% (Ours)   | 75.63 | 866 min |
>
> Notably, we remark that the above statistics could only give a rough estimation, because the actual running time depends on many factors unrelated to the algorithm itself, including how DataParallel is implemented, what supporting packages the codebase is using, et cetera. Statistics show that the RPG method is not significantly time-costly.
>
>
> Additionally, thank you for your advice on layouts and typos. We will amend them in later revisions.
>
> Sincerely,
>
> Authors
>
> References:
>
> [Chen et al.] Chasing sparsity in vision transformers: An end-to-end exploration. NeurIPS 2021.
>
> [Peste et al.] AC/DC: alternating compressed/decompressed training of deep neural networks. NeurIPS 2021.
>
> [Zhou et al.] Effective sparsification of neural networks with global sparsity constraint. CVPR 2021.

---

### Decision · Program_Chairs · 2023-09-21

**Decision:**

Accept (poster)

**Comment:**

This paper proposes a novel Rank-based PruninG (RPG) method for network pruning that maintains the ranks of sparse weights in an adversarial manner, leading to high-rank topology and improved performance. The proposed method is evaluated on various datasets and tasks, and compared to state-of-the-art methods. The experimental results show that the RPG method outperforms other pruning methods in terms of accuracy and efficiency. The main strengths of this paper are its novel combination of rank preservation and adversarial training ideas to address the limitations of existing pruning methods, the thorough experimental evaluation, and its clear presentation of results and analysis.

Reviewers initially raised several legitimate concerns including the hardware practical acceleration, the baseline inconsistency, and missing training time. The authors actively addressed the raised concerns during rebuttal by supplying many additional experiments and managed to convince even the most negative reviewer to turn positive. Eventually, based on the all-positive reviewer consensus, AC recommends acceptance.